# Physical Characteristics of Egg Yolk Granules and Effect on Their Functionality

**DOI:** 10.3390/foods12132531

**Published:** 2023-06-29

**Authors:** Beatrice Mofoluwaso Oladimeji, Ronald Gebhardt

**Affiliations:** Chair of Soft Matter Process Engineering (AVT.SMP), RWTH Aachen University, 52074 Aachen, Germany; ronald.gebhardt@avt.rwth-aachen.de

**Keywords:** egg yolk granules, macro-molecular protein structure, physical characteristics, functionality, applications

## Abstract

Eggs are among the most nutritious foods in the world, a versatile ingredient in many food applications due to their functional attributes such as foaming, emulsifying, and coloring agents. Many studies have been reported on egg yolk fractionation and characterization in the last decade because of its nutritional and health benefits, especially egg yolk granules. This has led to the development of new food products and packaging materials. However, the influence of their physical characteristics during processing significantly impacts the functionality of yolk granules. In this overview, the egg yolk, the granule fraction’s separation, fractionation, components, and molecular protein structure are first presented. Secondly, recent studies on egg yolk granules published over the past decade are discussed. Furthermore, the application of the granules in different industries and current specific scientific challenges are discussed. Finally, it simplifies the changes in the physical characteristics of the granules during different treatment methods and the impact on the functionalities of the resulting products in the food (emulsifiers, edible films), pharmaceutical, and health (encapsulation systems and biosensors) sectors.

## 1. Introduction

Eggs are recognized worldwide as an important part of the human diet because they are low in calories, inexpensive, and exceptional sources of complete protein. Both the egg white and egg yolk contain an equal distribution of protein yolk [1]; they provide appreciable vitamins, minerals, and trace elements, and possess multiple functionalities (foaming, gelation, emulsifying), making them excellent and versatile ingredients in food industry applications and home cooking. Eggs may vary in size and nutritional composition due to differences in their sources (hen, turkey, guinea, duck, quail, goose, pigeon, etc.) and rearing conditions [1].

Egg yolk is a natural colloidal dispersion [2] representing about 28–33% of the total egg weight [3,4] with high nutritional, biological, and functional values. It is a recommended complementary food for infants at the earliest age because of its high bioavailability [5], and it is a high-quality food emulsifier in processed products such as salad dressing, mayonnaise, baked goods, and gelling agents. Egg yolks are made up of water, lipid, and protein [5] and are tightly embraced by vitelline membranes which physically separate the yolk from the other egg compartments [1]. Egg yolks have a higher concentration of fat-soluble vitamins, essential fatty acids, and minerals [1] than egg whites. The protein fraction of egg yolk is considered more important in the food industry due to the excellent functional properties of its spherical lipoproteins [3]. This protein is composed of 68% low-density lipoproteins (LDLs), 16% high-density lipoproteins (HDLs), 10% globular proteins (livetins), 4% phosphoprotein (phosvitin), and 2% other minor proteins [6]. LDLs are the most abundant protein in egg yolk, providing egg yolk with emulsifying properties; however, HDLs, a macromolecular complex of proteins (75–80%) and lipids (20–25%) [7], have attracted broader attention, because they influence gel formation in egg yolks and can counterattack arteriosclerosis [8]. Egg yolks can be easily separated by centrifugation into a supernatant (80% water-soluble plasma fraction) and 20% insoluble pellets called granules [3,6]. The plasma mainly comprises 85% LDLs and 15% globular glycoproteins (α-, β- and γ-livetin), while the granules comprise 70% HDLs and 16% phosvitin linked by phosphocalcic bridges [9]. Egg yolk plasma exhibits better solubility due to its high content of floatable LDLs [9]; however, the granules contain HDLs that are considered good cholesterol because they protect against cardiovascular diseases. 

Egg yolk granules are a novel food, possessing many interesting characteristics (low cholesterol content, good emulsifying properties, or heat resistance) which aid their utilization as an ingredient in food applications [10]. The protein (phosvitin) found in egg yolk granules and its derivatives have also been demonstrated to possess anti-inflammatory, antioxidant, and anticancer activities and promote bone health [10,11]. Although the strong calcium phosphate bridge structure of the yolk granules is responsible for their low solubility, which is disadvantageous to their emulsifying functionality, destabilizing the egg yolk granules has proven to be an efficient strategy. Some of the many strategies are the application of pH and NaCl ionic strength [12] and high hydrostatic pressure [6], which improved the egg yolk granules’ solubility, the accessibility of its nutritional and bioactive compounds, and the digestibility and functionality of the product [6]. For many years, there have been interesting topics of study on egg yolk granules, but the limited study of their applications may be due to the method of destabilizing the egg yolk or its physical characteristics. Moreover, since egg yolk granules can yield different functional, biological, and nutritional properties when compared to egg yolk, there is a need to further investigate the potential to generate new ingredients from egg yolk granules for novel applications as foods, nutraceuticals, and polymers (microgels), considering the impact of the granule physical characteristics. This review investigated Elsevier’s Scopus Database based on the search query for “egg yolk” granule and structure*. A title–abstract–keyword search strategy was used, minimizing false negative results, and limiting documents to those published from 2001 to 2022. The review first presents brief up-to-date research studies on egg yolk based on the granule fraction. Secondly, it discusses the application of the granules and the current specific scientific challenges in the research area. Finally, the review clarifies the changes in the physical characteristics of the egg yolk granules during different treatments and the effects on their functionalities.

## 2. Extraction of Egg Yolk

For research, freshly laid eggs are usually obtained and manually broken. To separate the albumen from the vitellin membrane, the egg is carefully rolled on a blotting tissue to ensure no egg white proteins were mixed up. The vitellin membrane is then cut with a scalpel blade [7] or tweezer, and the yolk is collected.

Egg yolk is a natural emulsion made up of water, lipids (such as fats and oils), and proteins. The lipids and proteins are organized into various supramolecular structures within the emulsion, which give egg yolk its unique textural and functional characteristics, in different applications. The lipid component of egg yolk is primarily composed of phospholipids and triglycerides, while the protein component is primarily composed of various types of globular proteins such as ovalbumin, conalbumin, and livetin. These proteins can form complexes with lipids and other proteins, creating a hierarchical structure within the emulsion. Egg yolk constituents are difficult to separate due to their complex nature. Understanding the molecular and structural characteristics of each fraction of the egg yolk has clarified the approaches to be used in fractionating egg yolk [7]. Collected egg yolk is usually homogenized by stirring with a glass rod before fractionation into plasma and granules by dilution and centrifugation. Researchers have used this method extensively with modifications to the dilution conditions and centrifugation speed and time, which can define the attributes of the quality of the fractions obtained. Laca et al. [10] summarized a few methods of egg yolk production, which are similar in the basic steps of dilution and centrifugation. The dilution of the yolk was conducted by dispersing the yolk into an isotonic solution (NaCl) [7] followed by stirring, pH adjusting [13], and centrifugation. The precipitated granules were then washed and collected while the plasma fraction (supernatant) was repeatedly centrifuged for the complete removal of granules.

### 2.1. Egg Yolk Fractions and Constituents

Hen egg yolk’s main fractions, granules, and plasma contribute to the formation and stability of dispersed systems like emulsions by constituting an interfacial film. In egg yolk, high-density lipoproteins (HDLs) have a structure like globular proteins; single HDL proteins tend to form submicelles with a loop-like structure, like that of human HDL particles [7]. Additionally, low-density lipoprotein (LDL) is known as a main contributor to the emulsifying properties of egg yolk, but not much description is given about its molecular structure except that a mosaic structure exists at its surface [12]. In the plasma of egg yolk, the sizes of LDL micelles range from 17 to 60 nm with an average of 30 nm [7,12], while the particle size of the granule of egg yolk ranges from 900 nm at pH 7 [14] to 2000 nm [12] at extreme conditions.

Recent models of egg yolk and granule structures are shown in Figure 1. The first model depicts the structure of an egg yolk system described by Anton [12] revealing granules (the insoluble protein aggregates) suspended in the plasma (yellow fluid) containing LDL and soluble proteins. A model of egg yolk granules was proposed by Strixner et al. [7] in Figure 1B, revealing the pseudo-molecular structure of the HDL organized in a multi-level structure linked by phosvitin via calcium phosphate bridges and embedded LDL vesicles. Figure 1C describes an updated model of Figure 1A after the egg yolk was separated via mild centrifugation [15]. In Figure 1D, Naderi [15] shows the components of the microstructure and diameter of the egg yolk granules (0.84–4.87 µm), plasma LDL (20–60), and the model describing the refined molecular structure of granules before and after the phosphocalcic bridges were disrupted at an ionic strength above 0.25 NaCl. The author reported the homogeneity of different subfractions of the granule (Figure 1(Da)), while the dislocation of the phosphocalcic bridges occurs due to the solubility of the monovalent sodium substitute for divalent calcium ions. This causes the folate to link with the dissolved phosvitin, and HDLs to form a multi-level structure, held together by ionic forces (Figure 1(Db)).

### 2.2. Research Studies on Egg Yolk Processing, Applications, and Characterization

#### 2.2.1. Processing

Researchers have studied egg yolks and their fractions from different animals including hens, turkeys, ducks, etc. Many of the authors focused their research on exploring different methods of egg yolk fractionation and characterization and evaluating the structure of the protein and fractions using different methods and instruments. Laca et al. [13] described the lyophilization technique as a simple procedure for fractionating egg yolk, which will be solvent-free, with less denatured protein, good compound activity, and with increased fractions’ shelf life. Fresh and freeze-dried egg yolk fractions were characterized both physically and chemically, and the author concluded that the obtained granule fractions could be potential shelf-stable emulsifying or gelling agents suitable for use in food, particularly those requiring lower cholesterol content. Strixner and Kulozik [16] reported that heating egg yolk to 50 °C to a dry matter below 29% before the centrifugal fractionation with a moderate centrifugal acceleration of 5000× *g* resulted in excellent separation efficiencies. The study predicted this continuous fractionation process of liquid egg yolk is applicable even on the industrial scale level. 

Some authors investigated changes in the physicochemical, functional, and structural properties of egg yolk fractions as affected by different treatments including NaCl addition, soy lecithin addition, alkali and enzyme treatment, pH modification, phosphorylation, succinylation [17,18,19,20], and food ingredients [21]. The addition of NaCl increases protein solubility and surface hydrophobicity in the egg yolk compared to egg white dispersions [17]. The result could guide the processor through the application of egg white and egg yolk in the salt-containing egg gel-derived products. The physicochemical, functional, and structural properties of egg yolk treated with phospholipase A_2_, enzymes, and pH, were investigated while exploring a wide range of centrifugation separation parameters for optimal processing conditions [7,16,22,23]. Daimer and Kulozik [22] demonstrated that enzymatic (Phospholipase 2) treated egg yolk has higher heat stability and higher protein solubility, thereby resulting in improved emulsifying properties than untreated egg yolk. The authors later characterized oil–water emulsions prepared with the phospholipase 2 enzyme-treated egg yolk and concluded that they make a more effective emulsifier at a low pH than untreated egg yolk proteins. This is due to the enzyme’s ability to convert phospholipids into lysophospholipids, resulting in higher solubility and lower interfacial tension, hence enhancing the emulsifying properties [23].

Other researchers investigated the modification of egg yolk granules through the application of hydrostatic pressure. Xie et al. [5] investigated the effects of high-intensity ultrasound treatment on egg yolk, particularly on how this processing method could enhance its functional and structural properties. Additionally, the article of Sirvente et al. [24] reported some modified physicochemical characteristics of egg yolk, plasma, and granules pre-treated using a rotor-stator and high-pressure homogenization. This method caused active compounds that can adsorb at the interface of the oil–water emulsions to be produced from the pre-treated granules. In the long run, it has been reported that destabilizing the egg yolk granule via several mechanical treatments such as ultra-high-pressure homogenization (300 MPa pressure level, and 4 passes) has implications for the improvement of its digestibility, functionality, and access to its nutritional and bioactive compounds [6].

#### 2.2.2. Characterization and Applications

Research interest in this study field shifted to the application of egg yolk fractions and methods of improving the characteristics of the fractions. Egg yolk granules have found potential as ingredients in products such as Pickering emulsions, wheat dough, and muffins, as a low-cholesterol emulsifying agent in the preparation of mayonnaise [25], and edible films for food packaging options. The study of Rayner et al. [26] reported a useful application of egg yolk granules as Pickering emulsions, in the context of food and non-food formulations. Health-benefiting compounds (HDL) extracted from egg yolk have also been used in the preparation of Pickering emulsions [27]. Furthermore, Marcet et al. [28] studied the effect of egg yolk plasma and granules on the rheological and physical properties of batter and muffin recipes. The authors concluded that the substitution of 50% of the whole egg yolk by egg yolk granules is possible without significantly changing the physical properties of the muffin while substituting with the plasma fraction gives the muffin the textural and shape properties required. It also confirmed that the granular fraction of the egg yolk has significant nutritional value when compared to egg yolk plasma, and changes in the rheological and physical parameters are predictable with varying plasma/granules ratios included in the recipe. Moreover, the composition of the egg yolk granules was modified using a non-toxic procedure (increased ionic strength and power ultrasound), which improved the folate (5-MTHF) concentration up to 21 μg/g at 0.15 M NaCl [29]. The author later characterized the protein profile and microstructure of high-hydrostatic-pressure (HHP)-treated egg yolk [30]. The treatment led to the disintegration of the granule’s compact structure, but the folate was stable under 600 MPa at 5 min of treatment. There is also evidence of high concentrations of folate in the plasma fractions (230 µg/g dry matter) separated from the HHP-treated granule. Additionally, Fuertes et al. [31] developed edible protein films using egg yolk and fractions as raw materials and evaluated their mechanical and thermal properties. The author pointed out that the developed films are biodegradable and biocompatible and exhibit nutritional and appealing characteristics, thus making them valuable products for the food industry. In the work of Marcet et al. [32], the egg yolk granule fraction was also used to create good-quality, transparent, edible films after being pre-treated (at 45 °C for 40 min) using high-intensity ultrasound. Analyses conducted on the films revealed that the process improved the puncture strength of the film, the color remained unaffected, and the films had low water solubility. 

Leftover egg yolk granules (a by-product after phosvitin extraction) can find application in food products, as they had significant protein, rheological, and functional properties [33]. The determination of the stability of products that included egg yolk granules as ingredients under different NaCl ionic strengths, pH, and varying protein concentrations is another interesting area of research. It has been reported that egg yolk produces numerous lipid peroxidation products during processing and storage, and this could alter protein structural changes [34], and thus the quality of egg yolk and the product developed from it. The latest report on this field of study explored the formation and characterization of emulsion gel from heat-induced egg yolk granules–sodium alginate and then determined the in vitro digestibility and stability of β-carotene encapsulated in the emulsion gel. This was aimed to provide insight that egg yolk granules could find applications for nutrient delivery systems in food gels [35]. 

### 2.3. Scientific Recommendations from Egg Yolk Granules Research

From the documents studied, this review could recommend some further work to compare the impacts of the destabilization methods of egg yolk granules on their functional properties, mineral composition, protein, folic acid, proteomics, metabolomics, and microstructure. An understanding of the impact of structural modification on the functional properties of ultra-high-pressure homogenization-treated egg yolk granules [6] has been suggested in the literature. Further investigations on the microstructure of egg yolk granules will also provide a better understanding of the association between nutritional components (such as 5-MTHF) and granular protein composition [29], and the encapsulation approaches of important micronutrients for food fortification, supplements, and nutraceutical purposes. 

Many products in the food and pharmaceutical industries are processed at high temperatures (such as sterilization) and high pressure. There is also a need to further study and clarify the aggregation behavior, composition, structural, textural, and functional characteristics of the egg yolk fractions and their applicable products, as well as their composition and shelf stability as affected by high-heat processing conditions. This might provide some insights into key influencing factors that could cause changes in the properties of the egg yolk fraction and products when heat treatment is induced.

Individual components of egg yolk can be further characterized for functionality to find practical applications in various sectors. Egg yolk granules have found usefulness in bakery products (muffins) in replacing up to 50% of egg yolk due to its significantly reduced cholesterol and fat content [28], without significantly altering the physical properties of the final product. However, some scientific concerns have been raised regarding some textural and color differences observed in the developed products. There is, therefore, a need for further work to overcome noticeable differences in the rheological behavior of the batter which could negatively affect the sensory attributes of the final product due to interactions promoted by these granular components. Researchers could also work with egg yolk granule ingredients in other bakery foods such as cookies, bread, etc., and other food categories such as cereals, pasta, etc., to develop more value-added products with acceptable textural and sensory properties. Likewise, the report from the study of Mi et al. [36] on the use of egg yolk granules as emulsifiers in the preparation of high internal phase emulsions (gel-like concentrated emulsions), has raised a research need for exploring egg yolk granules in the transformation of oil from its conventional liquid-like to solid-like characteristics. This will facilitate the need to enrich food processing by providing a healthier option for consumers of oil and improving the storage stability of oils. Edible films from egg yolk granules can also be compared with those from other soft matter materials for solubility (acidic, alkaline, and neutral pH solution), better barrier properties, improved water resistance, and better mechanical properties. 

Further research can be carried out on the extraction, characterization, and utilization of value-added components of egg yolk granules as functional ingredients. This could help to efficiently utilize by-products or leftover egg yolk granules. Phosvitin (the major egg yolk protein), high-density lipoproteins, and IgY (egg yolk immunoglobin—an antibody), for example, can find useful applications in improving food and health potential. Furthermore, there is a need to investigate the capacity of egg yolk granules in food models as carriers of health-promoting components and as a delivery system of bioactive compounds in a simulated gastrointestinal tract. The efficiency of the microparticles employed for the encapsulation of polyphenols, probiotics, prebiotics, antioxidants, antimicrobial compounds, bioactive proteins, and peptides for nutraceutical and pharmaceutical uses might be investigated. The heat resistance of the bioactive compounds encapsulated using the egg yolk granules, as well as their stability during processing, dilution, and storage, need to be investigated or improved. The nutrient delivery efficiency of the microparticles, as affected by different chemical crosslinking and polysaccharide addition, can also be examined. The recent findings from the study of Li et al. [37] on the bioactivity and bioavailability of encapsulated curcumin present a challenging basis to further research the use of egg yolk granules as delivery systems for application in foods, supplements, and pharmaceuticals. 

## 3. Physical Characteristics of Egg Yolk Granules

### 3.1. Fractionation of Egg Yolk Granules and Their Composition

The extraction and purification of egg yolk constituents prove a bit challenging due to their complex composition. However, simple dilution and centrifugation methods have been used in the fractionation of the continuous aqueous phase (plasma) and insoluble denser structures (granules). The granules are the precipitated part of egg yolk after dilution and centrifugation, and the influence of its internal structure, constituents, and particle sizes are significantly affected after passing through different treatments. 

In the past few years, a variety of methods have been developed for the separation of egg yolk constituents. These methods include the use of organic solvents, which facilitate the extraction of proteins and the removal of lipids from the egg yolk [12], and enhance the solubility of the granules. An organic solvent resulted in an excellent extraction yield and purification rate, but its usage could be costly, time-consuming, and limited in some industrial sectors such as food and pharmaceuticals, due to harsh extraction efficiency and the possibility of solvent residuals in the final product. This is why some authors have explored the use of green techniques such as ultrasound, thermal, and high-pressure-assisted extraction methods. Table 1 summarizes some of these techniques and the composition of the granules separated. 

Lei and Wu [38] combined magnetic stirring with centrifugation after diluting the egg yolk with distilled water during the fractionation of egg yolk granules. The use of a magnetic stirrer is a quiet and more efficient method of extracting compounds but is limited to laboratory or small-scale experiments. Pasteurization (a mild-heat treatment in food) and high hydrostatic pressure (a non-thermal technology) are treatments known to reduce the microbial load in various food products and are also employed in egg yolk fractionation. HHP has a higher advantage as it does not involve heat intervention, allowing for the preservation of the nutritional and organoleptic properties of treated foods. Naderi et al. [39] confirm that the pasteurization and high hydrostatic pressure processing of egg yolk did not change its composition; however, the dilution of egg yolk before pressure treatment could significantly change the composition of the egg yolk fraction after centrifugation. Before dilution, the granule was 53.75% (dry matter basis—DB), and different dilution concentrations of the granule resulted in different values. The composition of the granule, protein, and folate ranged from 31 to 50%, 13 to 65%, and 1902.5 µg/100 g, respectively, for HHP-treated granules, and from 42 to 47%, 45 to 63%, and 342–1763 µg/100 g for the granule, protein, and folate contents, respectively, in pasteurized granules.

Contrary to freezing (a non-thermal process) where the separation of granules is not possible from the egg yolk, the pasteurized and HHP-treated granules allowed a full recovery of the folate in the granules. The pasteurization and centrifugation of the egg yolk at a lower centrifugal force promoted fractionation [40] but resulted in lower granule (31.8%) recovery. This could be the reason why many researchers make use of a higher centrifugal force of 10,000× *g*, as it has been reported that it allows for the complete separation of granules and plasma [16]. The use of ultra-high-pressure homogenization (UHPH), an innovative simultaneous sterilization and homogenization technology, is also possible in egg constituent separation. The study of Gaillard et al. [6] employed UHPH technology; this treatment modified the structure of egg yolk granules and reduced the particle size distribution. However, increasing the pressure to 300 MPa enhanced larger particles through protein–protein interaction. Lyophilization (freeze-drying) is a water removal process used to extend the shelf life of food. Egg yolk freeze-dried post-dilution and -centrifugation allowed for the separation of the granules [13] and proved capable to extend the shelf life of the fraction.

Different studies have employed the pre-mentioned procedure with some modifications in the yolk dilution factor, dilution medium (aside from distilled water), and/or centrifugation conditions. This includes dilution in salt solutions (such as NaCl, (NH_4_)_2_SO_4_) before extraction using a magnetic stirrer and ultrasonic-assisted treatment, to change the ionic strength, adjust the pH [16] for a better disintegration of aggregated granules into small particles and micelles [36], and also enhance phosvitin separation [41]. Other researchers employed alternatives such as a phosphate buffer or deionized water during dilution [10]. Increasing the volume of water used in dilution increases the lipid (lipoproteins) content in the granules [38], this explains why the amount of precipitate after centrifugation increased with increasing water dilution [38]. An equal volume of water dilution in egg yolk granules is comparable to dilution in 1% (0.17 M) NaCl and generates dry granules with minimum lipid content. This means that the dilution of egg yolk in an equal volume of water before centrifugation could be a more cost-effective and acceptable dilution method than the use of salts. Based on this research study, many studies on egg yolk fractionation were performed by employing a higher centrifugal force (10,000× *g*) at a controlled temperature of 4 °C and for long residence times (commonly 45 min). The easy and gentle method of separation of egg yolk fractions has proven to provide high separation efficiencies. Other factors influencing the continuous centrifugal separation of egg yolk granules were described by Strixner and Kulozik [16]. The application of egg yolk granules is becoming increasingly interesting; there is, therefore, a need to increase its value addition by developing simple, cost-effective, yielding, efficient, and reproducible methodologies for its protein separation, which can also be adapted to the pilot scale level. 

### 3.2. Physical Treatments of Egg Yolk Granules and Parameters Determination

This sub-section summarizes physical treatments of egg yolk granules, the objective behind the treatment, highlighted results, and possible effects on its functionality, as described in Table 2. Mechanical treatments such as high-intensity ultrasonic (HIU) could improve the functionality (emulsification, foaming, and gel properties) of egg yolk granules by altering the aggregation of the egg yolk components and the partial dissociation of the calcium phosphate bridge, which enables particle size reduction [5]. In the study of Naderi et al. [39] HHP treatment did not modify the folate or the lipid composition of the granules, allowing maximum recovery of the folate content. Gaillard et al. [6] also revealed that UHPH treatment seems severe to the granule structure but the phosvitin–HDL complex showed very high resistance, and their composition was unaffected by the process. The treatment of granule fractions at 400 MPa for 5 min has proven effective in the extraction of the water-soluble B vitamin (5-MTHF) using high hydrostatic pressure. This shows that under mechanical treatment, the functionality of the treated egg yolk granules is improved.

Heat treatment such as pasteurization is the most used treatment method in egg production, which is why some researchers also focus on their influence on the functional properties of the egg yolk fractions. Importantly, egg yolk granules can withstand heat treatment more than other fractions [43] up to pasteurization temperature, and still retain good functional (emulsifying) properties [44]. The report of Naderi et al. [39] confirmed that pasteurization (61 °C) of the egg yolk granules had no significant impact on their composition, but allowed the full recovery of the folate concentration into the granule fraction. This shows that egg yolk granules have high heat resistance, and there is little or no effect on their functionality under milder and short thermal treatment. This was not the case in freezing treatment where folate concentration decreased significantly by 25% due to possible degradation during freezing. Furthermore, phosvitin could be separated using thermal treatment [45] in a water bath at varying temperatures, but with a lower recovery rate compared to other methods. However, the treatment method was reported to be cost-effective and sustainable, producing phosvitin with a very high ferrous and ferric iron-binding ability that could find excellent applications as an important ingredient in several industries. As summarized in Figure 2, mechanical and thermal treatment induces positive changes to the functionality of egg yolk granules, in terms of sources of health-benefiting components, emulsifying applications, films, and encapsulation properties. It is important to know that chemical characterization will also be affected, and this is considered crucial in their application. 

Table 3 reveals some physical parameters measured in egg yolk granules, which are targeted toward separation efficiency and microstructure determination. Sedimentation has been used in many processes to separate two liquids or a liquid and a solid, but not enough to accomplish separation efficiently, which is why the process is accelerated using heat or centrifugal force. The separation efficiency of egg yolk granules is generally calculated using the sedimented granule dry matter mass flow rate, expressed as a function of the *k*-factor. The separation efficiency of high-density lipoprotein (HDL) granules from hen egg yolk was investigated [7]. The authors reported that separation efficiency is dependent on pH, and the decrease is stronger at the isoelectric point of pH 4.0, resulting in an increased k-value. In addition, the author revealed that increasing the centrifugal acceleration does not imply an increased separation efficiency, as a moderate acceleration of 5000× *g* enabled complete granule fraction removal. The study of Strixner and Kulozik [16] also found that heating the egg yolk to 50 °C and reducing dry matter content below 29% resulted in excellent separation efficiencies of the granule fraction. It was reported that limiting the dry matter concentration to 29% would allow sedimentation within the polydisperse particle at the same velocity. 

The microstructure of egg yolk granules has been measured using different techniques, and the conditions at which the granule is prepared (varying chemical compositions) result in changes in the structure of the granules [7]. Strixner et al. [12] determined the granules’ structural knowledge using both atomic force microscopy (AFM) and surface-sensitive X-ray scattering. The result from the AFM images revealed that the HDL granules are strongly dependent on the pH of the solution, and structural changes were difficult to detect due to variations in the chemical composition on the sample surface, and the pH of the solution. Using X-ray scattering in reflection geometry, two prominent lateral lengths of the granular structure in the sample surface could be resolved. These are shortened at the isoelectric point near pH 4. The reduced repulsive forces under these pH conditions lead to a more compact structure that can more effectively incorporate LDL vesicles. Furthermore, the use of scanning electron microscopy (SEM) revealed close packing structures of protein spheres in fresh and lyophilized granules, and a well-organized structure of the coagulated granule fraction, which is close to that of the global protein due to the high HDL granule content [13]. Valverde et al. [46] observed the spherical structure of granule gels using SEM and confirmed their particle size corresponds with HDL–phosvitin complexes. 

Physical treatment of egg yolk granules using calcium chelators is also possible. Fu et al. [14] reported the use of sodium tripolyphosphate (Na_5_O_10_P_3_), ethylenediaminetetraacetic acid disodium (EDTA-2Na), and trisodium citrate (C_6_H_5_Na_3_O_7_) in dissociating micro-sized granules into nanoparticles with good colloidal stability at pH 7.0. This dissociation did not change the chemical bonds or protein profiles of the granules but improved their functionality and storage stability in an emulsion.

## 4. Conclusions

Egg yolks have immense potential and have gained much attention in the past decade. This study summarizes the current state of knowledge on the structure, processing, and functionality as well as the changes in the physical characterization of the granules during different treatment methods. As discussed above, methods to destabilize granules have been researched, many of which have some unacceptable effects (such as a high concentration of NaCl’s effect on flavor and health benefits), energy-intensive effects, and so on. This is why some authors are exploring the use of traditional thermal mechanical treatments or non-thermal emerging technologies to improve the separation, functionality, and stability of egg yolk granules. It has great natural anti-obesity, atheroprotective, and anti-osteoporosis potential in the functional food, food packaging, pharmaceutical, health, mechanical, and sustainable environmental fields. This review could provide an understanding of the content of the research area and a valuable strategy for potential research to characterize the bioactive components of the granules, develop new functional materials such as responsive encapsulation systems or biosensors, and search for applications for them in the food, pharmaceutical, and healthcare sectors. As the future need for nutritious, healthy, safe, and sustainable products is pursued worldwide, improving the functionality of egg yolk granules through physical treatment or processing is a huge step toward expanding this scope. 

## Figures and Tables

**Figure 1 foods-12-02531-f001:**
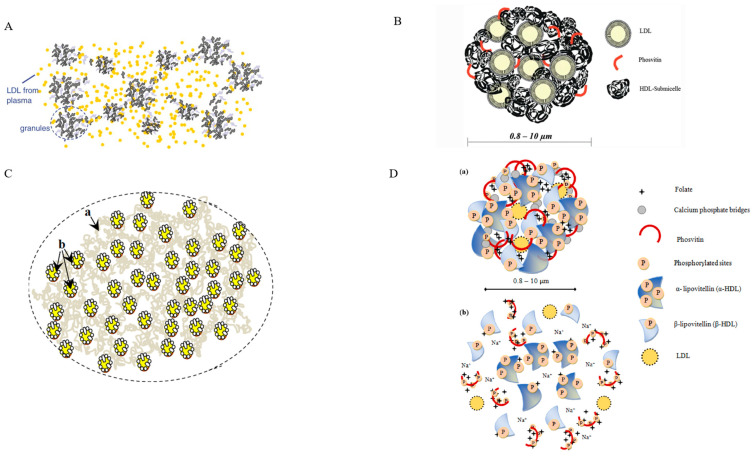
(**A**) Structure of egg yolk system by [12]; (**B**) schematic model of egg yolk granule proposed by [7]; (**C**) components of the microstructure of egg yolk: (**a**) granule and (**b**) LDL in plasma proposed by [15]; and (**D**) representation of the model of (**a**) native and (**b**) disrupted egg yolk structure proposed by [15].

**Figure 2 foods-12-02531-f002:**
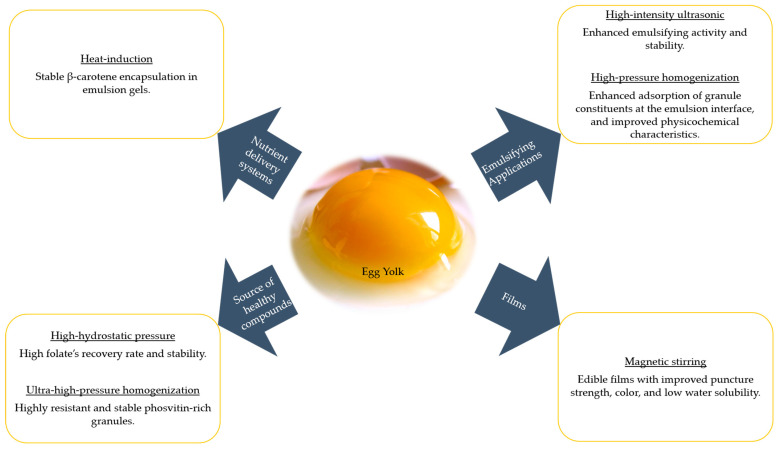
Diagram summarizing applications of current mechanically and thermal-treated egg yolk granules.

**Table 1 foods-12-02531-t001:** Egg yolk granule constituents based on the methods of fractionation.

Pre-Treatment	Dilution Ratio (Yolk:dH_2_0)	Centrifugation Conditions	Granule Constituents	Conclusion	References
Mechanical					
Magnetic stirring (1 h at room temperature)	1:1	10,000× *g*/ 45 min/4 °C	Lipids (5.8–53.8%) Phosvitin (14.7%)	Increasing water dilution volume increased dry granules containing a minimum amount of lipids.	[38]
High hydrostatic pressure (HHP) (400 MPa for 5 min)	Different concentrations (0.1, 1.0, 10, 25, and 50%)	10,000× *g*/ 45 min/4 °C	Granule (53.57%) Protein (70.27%) Folate (1902.5 µg/100 g)	Dilution before HPP treatment changed the composition of the granule. Different ratios resulted in different compositions, but the 50% ratio maintained the original compositional characteristics and enhanced folate recovery.	[39]
Ultra-high-pressure homogenization (50–300 MPa, 1–4 passes)	1:1 (*w*/*v*)	10,000× *g*/ 45 min/4 °C	Granule (99.1%) Protein (62.8%) Lipid (42.0%)	Restructuration of granules, but approximate compositions were not modified. The average particle size decreased, and the mean diameter of particles increased at 300 MPa; the value was highest at 4 passes.	[6]
Thermal					
Pasteurization (3.5 min at 61.1 °C)	1:1 *w*/*w*	10,000× *g*/ 45 min/4 °C	Granule (45.58%) Protein (66.03%) Lipid (24.84%) Folate (1969.7 µg/100 g)	Pasteurization did not alter the composition of the egg granule.	[39]
Pasteurization (2 min at 63 °C)	NM	2800× *g*/ 8 min/5 °C	Granule (31.8%) Protein (64.1%) Lipid (34.3%)	Pasteurization promoted better fractionation at lower acceleration.	[40]
Non-thermal					
Freezing (−18 °C)	Not Permitted	Not Permitted	Not determinable	Separation of granules is not possible, because of the treatment-induced denaturation of the sensitive lipid-protein complexes.	[39]
Freeze drying (−70 °C, 0.1 mBa)	1:1 *v*/*v*	10,000× *g*/ 45 min/4 °C	Granule (41.4%) Protein (24.0%) Lipid (16.6%)	The lyophilization process increased granule shelf life, with barely any negative effect on the product characteristics.	[13]

**Table 2 foods-12-02531-t002:** Some physical treatment/processing of egg yolk granules and implications.

Unit Operation	Objective	Condition	Highlighted Result	Conclusion	References	Comment on Functionality
Mechanical						
High hydrostatic pressure (HHP)	Effect on folate recovery	Pressure level (400 MPa for 5 min)	All folate concentrations recovered in granules	Highly stable folate	[39]	Composition not altered in terms of folate, protein, and lipid. Could be functional in various industries.
Ultra-high-pressure homogenization (UHPH)	Improve techno-functionality of egg yolk granule	Pressure levels (50, 175, and 300 MPa)	Microstructure altered, due to the large protein network formed.	Highly stable granules	[6,42]	Very-high-resistance phosvitin, unaffected proximate composition; improved water and oil binding capacities.
Thermal						
Pasteurization	Effect on folate recovery	Pasteurize at 61.1 °C for 3.5 min.	All folate concentrations recovered in granules	No significant impact on granule composition	[39]	Folate-enriched granules can find functionality in various industries.
Non-thermal						
Freezing	Effect on folate recovery	Freezing (−18 °C), thawing (4 °C for 30 h)	Fractionation is not possible; degradation occurred during freezing	Folate concentration decreased significantly by about 25%.	[39]	The procedure decreased folate concentration significantly. Reduced functionality due to irreversible loss of fluidity.

**Table 3 foods-12-02531-t003:** Measurement of egg yolk granules’ physical parameters.

Analytical Parameters	Pre-Treatment of Egg-Yolk Granule Prior/ Pro Fractionation	Condition of Granule Prior Analysis	Results	References
Sedimentation				
Separation efficiency	22% DM 4 °C, g-force: 2000–10,000× *g*	Granules resuspend in 0.15 M sodium chloride solution	Separation efficiency is strongly dependent on pH.	[7]
Separation efficiency	22% DM Heating to 50 °C	Granule: 0.15 M isotonic sodium chloride solution (1:2)	A much better separation efficiency of the granule was obtained.	[16]
Microstructure				
Atomic force microscopy	10,000× *g* centrifuged granule resuspended in 0.15 M sodium chloride solution	Adjusted to pH 4.0 using 1.0 M HCl	Structural changes are difficult to detect due to varying chemical compositions and pH.	[7]
X-ray scattering (Beam *α_i_* 0.89°), distance 2014 mm pixel size 172 µm	10,000× *g* centrifuged granule resuspended in 0.15 M sodium chloride solution	Adjusted to pH 4.0 using 1.0 M HCl	X-ray scattering reflects the geometry of the nanostructure and composition.	[7]
Scanning electron microscopy (SEM)	NaOH was added to adjust pH to 7	Encapsulated in 2.5% agar at 45 °C, Size: 3–4 mm cubes, Fixed: 3% glutaraldehyde in 25 mM phosphate buffer (pH 6.8) Post-fixed: 2% osmium tetraoxide with 0.1 M imidazole Dehydrated in ethanol Finally in 100% acetone	Close packing structures of protein spheres.	[13]
SEM	The pH of the egg yolk was adjusted to 7 using 1 N NaOH, and egg yolk granules lyophilized	Fixed: 3% glutaraldehyde in 25 mM phosphate buffer Dehydrated in ethanol Finally in 100% acetone	Spherical structures of granule gels within 0.3 and 2 µm.	[46]

## Data Availability

Data is contained within the article.

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
