# Peer review of "Physical Characteristics of Egg Yolk Granules and Effect on Their Functionality"

_foods, 2023, doi:10.3390/foods12132531_

Round 1

Reviewer 1 Report

I reviewed this manuscript very carefully. The topic of this review is interesting, but I regret to inform that the present form is not sufficient for publication in Foods.

1. The authors mention that one of the biggest challenges faced in egg yolk granule research is the effect of the physical characteristics during processing on its functionality. I suggest this review is focused on the changes of physical characteristics of egg yolk granules during different treatment methods and the effects on their functionality. And the most important point is how we can improve their functionality through physical treatment/processing and thus broaden the field of application. And their application is better to be classified. However, in this review, much known information on egg yolk and egg yolk granules has been reported, and the logic is not clear.

2. The physical treatment/processing will induce changes on the egg yolk granules’ physical parameters, but their chemical characterization will also be affected, which was also very crucial to their application.  

There are some grammatical errors and instances of badly worded/constructed sentences.

Author Response

General comment: I reviewed this manuscript very carefully. The topic of this review is interesting, but I regret to inform that the present form is not sufficient for publication in Foods. There are some grammatical errors and instances of badly worded/constructed sentences.

General response: Thank you for your sincere comment regarding our manuscript. We have improved the state of the manuscript to make it sufficient for publication in Foods. Grammatical errors have been corrected, and sentences reworded.

Point 1: The authors mention that one of the biggest challenges faced in egg yolk granule research is the effect of the physical characteristics during processing on its functionality. I suggest this review is focused on the changes of physical characteristics of egg yolk granules during different treatment methods and the effects on their functionality. And the most important point is how we can improve their functionality through physical treatment/processing and thus broaden the field of application. And their application is better to be classified. However, in this review, much known information on egg yolk and egg yolk granules has been reported, and the logic is not clear.

Response 1: The review is focused on the changes in physical characteristics of egg yolk during different treatments, and the effect on their functionality, as mentioned. This has been cleared across the manuscript, in lines 12-16, 17-20, 73-80, 465, 473-480. Their application has been improved and presented in Figure 2 (lines 408-409 ).

Point 2: The physical treatment/processing will induce changes in the egg yolk granules’ physical parameters, but their chemical characterization will also be affected, which was also very crucial to their application. 

Response 2: This statement has been added to the manuscript (lines 412-415)

Reviewer 2 Report

The paper is a good review of an interesting subject as is the study of the Egg yolk Granules and their applications and benefits. It is good presented and structured with good bibliography search of the latest work in the subject.

Author Response

Point 1:  The paper is a good review of an interesting subject as is the study of the Egg yolk Granules and their applications and benefits. It is good presented and structured with good bibliography search of the latest work in the subject.

Response 1: Thank you for considering our manuscript worthy of publication, and for taking the time to review it.

Reviewer 3 Report

This work presents a review about egg yolk granules and its applications in different industries and scientific challenges.

It is known that the egg yolk is a nutritious and important product, but this article does not present relevant results.

This topic is not original, as any researcher can find the results obtained here with a bibliographic search of the literature. In addition, it does not present the methodology used in this research, such as the keywords used, or even the definition of the period covered.

Furthermore, the authors do not present challenges and future trends in this article, topics commonly found in review papers.

In my opinion this article should be rejected.

Author Response

General comment: This work presents a review of egg yolk granules and its applications in different industries and scientific challenges. It is known that the egg yolk is a nutritious and important product, but this article does not present relevant results. This topic is not original, as any researcher can find the results obtained here with a bibliographic search of the literature. In addition, it does not present the methodology used in this research, such as the keywords used, or even the definition of the period covered.

Response 1: Thank you for the time taken to review our manuscript. Sorry, we didn’t meet up with your expectations. We have revised the manuscript to make it worthy of publication, based on your comments and that of the other reviewers and editor. We have also put together this review using results obtainable from the bibliographic search of the literature, giving future researchers an overview of the research field and the results obtained.

Point 1: This work presents a review of egg yolk granules and its applications in different industries and scientific challenges. It is known that the egg yolk is a nutritious and important product, but this article does not present relevant results. This topic is not original, as any researcher can find the results obtained here with a bibliographic search of the literature. In addition, it does not present the methodology used in this research, such as the keywords used, or even the definition of the period covered.

Response 1: We have investigated trends in the research of egg yolk granules, searching Elsevier’s Scopus database. This information has been included in the manuscript (lines 73-76). Findings from the search and reported by authors in the literature are included in the manuscript.

 Point 2: Furthermore, the authors do not present challenges and future trends in this article, topics commonly found in review papers.

Response 2: Current scientific challenges found in the reviewed articles are presented in the manuscript.

Round 2

Reviewer 3 Report

The authors carried out a good review of the article and it was much better explained. They included the methodology used in the research, such as the use of the Scopus platform, keywords, period and the search strategy title- abstract-keywords. In addition, some future challenges and trends have been included.